# Impact of Antibiotic Prophylaxis Duration on the Incidence of Healthcare-Associated Infections in Elective Colorectal Surgery

**DOI:** 10.3390/antibiotics14080791

**Published:** 2025-08-04

**Authors:** Vladimir Nikolic, Ljiljana Markovic-Denic, Boris Tadić, Milan Veselinović, Ivan Palibrk, Milorad Reljić, Predrag Sabljak, Lidija Masic, Svetozar Mijuskovic, Stefan Kmezic, Djordje Knezevic, Slavenko Ostojić, Jelena Đokić-Kovač, Andrija Antic

**Affiliations:** 1Institute of Epidemiology, Faculty of Medicine, University of Belgrade, Visegradska 26a, 11129 Belgrade, Serbia; vladimir.nikolic@med.bg.ac.rs; 2Department for HPB Surgery, Clinic for Digestive Surgery, University Clinical Centre of Serbia, Koste Todorovica No. 6, 11000 Belgrade, Serbia; tadicboris@yahoo.com; 3Department for Surgery with Anesthesiology, Faculty of Medicine, University of Belgrade, Dr Subotica No. 8, 11000 Belgrade, Serbia; milan.veselinovich@gmail.com (M.V.); ivanpalibrk@yahoo.com (I.P.); predragsabljak63@gmail.com (P.S.); kstefan1986@gmail.com (S.K.); dr.djordje.knezevic@gmail.com (D.K.); slavenko.ostojic@kcs.ac.rs (S.O.); jelenadjokickovac@gmail.com (A.A.); 4Clinic for Digestive Surgery, University Clinical Center of Serbia, Koste Todorovica No. 6, 11000 Belgrade, Serbia; misoreljic@gmail.com (M.R.); lidijaamed@gmail.com (L.M.); 5Department of Anesthesiology, Reanimatology and Intensive Care, Clinic for Digestive Surgery, University Clinical Centre of Serbia, Koste Todorovica No. 6, 11000 Belgrade, Serbia; 6Faculty of Medicine, University of Belgrade, 11129 Belgrade, Serbia; svetozarmijuskovic3@gmail.com (S.M.); drandrija.antic@gmail.com (J.Đ.-K.); 7Center for Radiology and Magnetic Resonance Imaging, University Clinical Center of Serbia, Pasterova No. 2, 11000 Belgrade, Serbia

**Keywords:** antibiotic prophylaxis, colorectal surgery, healthcare-associated infections

## Abstract

**Background/Objectives**: Antibiotic prophylaxis is a key component of infection prevention strategies. This study aimed to evaluate whether the duration of antibiotic prophylaxis influences the incidence of HAIs in patients undergoing elective colorectal surgery. **Methods**: This prospective cohort study included 278 adult patients who underwent elective colorectal surgery at a single tertiary care center. Patients were categorized into two groups based on the duration of antibiotic prophylaxis: one day or more than one day. Data on demographics, clinical characteristics, perioperative variables, and infection outcomes were collected. **Results**: The overall incidence of HAIs was 16.9%, with no significant difference between patients receiving one-day versus extended antibiotic prophylaxis. However, traditional multivariate analysis showed that prophylaxis lasting more than one day was independently associated with a significantly lower risk of HAI (RR = 0.30, 95% CI: 0.12–0.75, *p* = 0.010) and surgical site infections (RR = 0.24, 95% CI: 0.08–0.72, *p* = 0.011). After adjusting for confounders using propensity score matching, this association was no longer statistically significant. No significant association was found between prophylaxis duration and urinary tract infections. Regarding antibiotic selection, first-generation cephalosporins were the most commonly used agents, accounting for 78.8% of prophylactic prescriptions. This was followed by fluoroquinolones (14.4%) and third-generation cephalosporins (5.0%). All patients received metronidazole, a nitroimidazole-class antimicrobial, in combination with the above agents. **Conclusions**: One day of prophylactic antibiotics may be sufficient in SSI prevention in patients undergoing elective colorectal surgery. The use of extended antibiotic prophylaxis beyond one day should be considered for high-risk patients at high risk of infection, particularly those requiring ICU care.

## 1. Introduction

Healthcare-associated infections (HAIs) remain one of the most common and serious complications of surgical procedures, particularly in colorectal surgery, where the risk is elevated due to the high bacterial load of the large intestine and the complexity of the interventions involved [1,2]. HAIs in this population contribute to prolonged hospital stays, increased healthcare costs, and higher postoperative morbidity and mortality [3].

Antibiotic prophylaxis is a basis of infection prevention in colorectal surgery. International guidelines, including those from the World Health Organization (WHO), the Centers for Disease Control and Prevention (CDC), and the American Society of Health-System Pharmacists (ASHP), recommend administering prophylactic antibiotics within 60 min before incision and discontinuing them within 24 h after surgery for clean-contaminated procedures [4,5]. However, real-world practice frequently diverges from these recommendations. Surgeons often extend antibiotic prophylaxis beyond 24 h, particularly in high-risk patients and due to concerns regarding infection control [6].

Despite the widespread use of prolonged regimens, the evidence supporting their effectiveness remains inconsistent. Some studies have shown no benefit or even harm from extended prophylaxis due to increased risk of antimicrobial resistance and adverse effects [7].

The objective of this study was to evaluate whether the duration of antibiotic prophylaxis is associated with the incidence of healthcare-associated infections in patients undergoing elective open colorectal surgery.

## 2. Results

A total of 278 patients were included in the analysis, of whom 99 (35.6%) received antibiotic prophylaxis for one day and 179 (64.4%) received antibiotic prophylaxis for more than one day (Table 1). No statistically significant differences were observed between the two groups in terms of sex distribution (*p* = 0.579), age (*p* = 0.386), Charlson comorbidity index (*p* = 0.771), wound classification (*p* = 0.684), smoking status (*p* = 0.242), alcohol consumption (*p* = 0.299), radiotherapy (*p* = 0.801), chemotherapy (*p* = 0.913), number of staff in the operating room (*p* = 0.771), intensive care unit (ICU) stay duration (*p* = 0.343), or length of hospitalization before surgery (*p* = 0.078). A statistically significant difference between the groups was observed in body mass index (BMI) (*p* = 0.021), with higher proportions of overweight and obese patients in the group that received prophylaxis for more than one day. Additionally, patients in this group had a higher number of drains (*p* < 0.001), a longer duration of drain use (*p* < 0.001), longer length of urinary catheter use (*p* < 0.001), and were significantly more likely to be admitted to the ICU postoperatively (60.9% vs. 16.2%, *p* < 0.001). Regarding American Society of Anesthesiology (ASA) classification, a higher proportion of patients in the one-day group had ASA scores of 1–2 (57.7% vs. 45.3%), while scores of 3–4 were more frequent in the >one-day group (54.7% vs. 42.3%; *p* = 0.048).

As shown in Table 2, the overall incidence of healthcare-associated infections was 16.9%, with no significant difference between the one-day and extended prophylaxis groups (17.2% vs. 16.8%, *p* = 0.930). The incidence of surgical site infections was 9.7% (11.1% vs. 8.9%, *p* = 0.558), urinary tract infections 6.8% (6.1% vs. 7.3%, *p* = 0.704), and Clostridioides difficile infections 1.1% (2.0% vs. 0.6%, *p* = 0.289), with no significant differences between groups. However, bloodstream infections occurred only in the group with extended prophylaxis (3.9%), with a borderline statistically significant difference (*p* = 0.053). The occurrence of infections of decubitus ulcers was rare and equally distributed (*p* = 1.000). Superficial SSIs were observed in 7.1% of patients in the one-day prophylaxis group and 5.6% in the >one-day group. Deep SSIs occurred in 2.0% and 2.2% of patients. Organ-space SSIs were rare, with incidences below 2% in both groups. Distribution of SSI types according to the duration of the antibiotic showed no statistical significance (*p* = 0.957). There was a statistically significant difference in the rates of HAIs based on ICU admission status (*p* < 0.001). Among the 47 HAIs, 33 cases (70.2%) occurred in patients who had been admitted to the ICU.

The results of traditional multivariate binary logistic regression models for predicting HAIs before and after PSM are presented in Table 3. Before PSM, extended antibiotic prophylaxis (>1 day) was independently associated with a significantly lower risk of HAIs (aRR 0.33, 95% CI 0.14–0.78, *p* = 0.012). However, after applying PSM to adjust for potential confounding by indication, this association was no longer statistically significant (aRR 0.81, 95% CI 0.33–1.98, *p* = 0.636).

Following PSM, higher ASA scores (aRR 2.54, 95% CI 1.08–6.01, *p* = 0.033), a longer duration of drain use (aRR 1.11 per day, 95% CI 1.00–1.22, *p* = 0.046), worse wound classification (aRR 22.0, 95% CI 3.78–128.0, *p* < 0.001), and ICU admission (aRR 4.51, 95% CI 1.74–11.69, *p* = 0.002) were identified as significant independent predictors of HAIs.

Table 4 presents the multivariate binary logistic regression models for predicting SSIs before and after PSM. Prior to PSM, extended prophylaxis (>1 day) was significantly associated with a reduced risk of SSI (aRR 0.26, 95% CI 0.09–0.75, *p* = 0.013). However, after adjusting for potential confounding through PSM, this association was no longer statistically significant (aRR 0.48, 95% CI 0.17–1.36, *p* = 0.167).

In the post-PSM model, higher ASA scores (aRR 3.61, 95% CI 1.26–10.32, *p* = 0.017), a longer duration of drain use (aRR 1.14 per day, 95% CI 1.01–1.28, *p* = 0.030), and ICU admission (aRR 5.34, 95% CI 1.70–16.83, *p* = 0.004) remained significant independent predictors of SSI.

Table 5 presents the multivariate binary logistic regression models for predicting UTIs before and after PSM. In the pre-PSM model, extended prophylaxis (>1 day) was not significantly associated with UTI risk (aRR 0.71, 95% CI 0.19–2.66, *p* = 0.612). After PSM, this association remained non-significant (aRR 2.76, 95% CI 0.52–14.70, *p* = 0.233).

Following PSM, significant predictors of UTIs included a longer duration of urinary catheter use (aRR 1.11 per day, 95% CI 1.03–1.20, *p* = 0.005) and contaminated wound classification compared to clean-contaminated wounds (aRR 16.2, 95% CI 3.69–71.36, *p* < 0.001).

Surgical type distribution did not significantly differ between patients receiving one-day and extended antibiotic prophylaxis. Among those receiving one-day prophylaxis, 34.3% underwent colon surgery and 65.7% rectal surgery, whereas in the >1-day group, 38.0% had colon surgery and 58.1% rectal surgery (*p* = 0.095).

Regarding antibiotic selection, first-generation cephalosporins were the most commonly used agents, accounting for 78.8% of prophylactic prescriptions. This was followed by fluoroquinolones (14.4%) and third-generation cephalosporins (5.0%). Other classes, including second-generation cephalosporins, penicillins with antipseudomonal activity, carbapenems, and fourth-generation cephalosporins, were used infrequently. All patients received metronidazole, a nitroimidazole-class antimicrobial, in combination with the above agents.

The duration of antibiotic prophylaxis among all patients ranged from 1 to 11 days. The mean duration was 3.17 ± 2.62 days, while the median was 2.0 days. Most patients received between 1 and 3 days of prophylaxis, with a small number receiving prolonged regimens up to 11 days.

The duration of antibiotic prophylaxis varied slightly by antibiotic class without statistical significance (*p* = 0.376). Patients receiving third-generation cephalosporins had the longest mean duration of prophylaxis (3.86 ± 2.48 days), followed by those receiving penicillin with antipseudomonal activity (3.50 ± 0.71 days), fluoroquinolones (3.28 ± 2.66 days), and first-generation cephalosporins (3.11 ± 2.65 days). In all groups, the median duration was 2–3 days.

Out of a total of 27 SSIs, microbiological confirmation of the causative agent was obtained in 13 cases (48.1%). The most frequently microorganisms isolated in the surgical sites were *Enterococcus* spp. (46.1% of cases) and bacteria from the family *Enterobacteriales*, *Klebsiella* spp., *E. coli,* and *Pseudomonas aeruginosa*. In patients with laboratory-confirmed UTIs, the most frequently isolated pathogens belonged to the *Klebsiella* spp. group, identified in 42.9% of cases, indicating the predominance of Gram-negative bacilli in the etiology of UTIs in this population.

Blood cultures yielded two strains of *Enterococcus faecium*, *Pseudomonas aeruginosa*, and *Klebsiella pneumoniae* and one strain of *S. aureus*. The causative agents of pressure ulcer infections were *Acinetobacter* spp. (resistant to carbapenems), two isolates of *Pseudomonas aeruginosa* (one sensitive and one resistant to carbapenems), *Klebsiella* spp. (resistant to third-generation cephalosporins and with intermediate sensitivity to carbapenems), and *Enterococcus* spp. (resistant to glycopeptides). None of the identified pathogens were resistant to all antibiotics. Resistance patterns are presented in Table 6.

## 3. Discussion

This study aimed to investigate the relationship between the duration of antibiotic prophylaxis and the incidence of healthcare-associated infections (HAIs) in patients undergoing colorectal surgery by comparing a one-day prophylaxis antibiotics group with an extended prophylaxis group.

The incidence rate of all HAIs in this study was approximately 17%, which is higher than rates in developed countries [8]. Postoperative infections are more common in colorectal surgery than in other types of surgery due to the potential bacterial load from the colon and rectum [9]. The most frequent HAI types in surgery in general, as well as in colorectal surgery, are SSIs. The SSI rates in colorectal surgery range from 5% to 30% [10,11], even to 60% in low- to middle-income countries [12], although rates have been decreasing in recent years [13]. The SSI incidence in our study was 9.7% and there were no differences between the two groups regarding all SSIs. However, the superficial incision SSI rate was lower in the extended prophylaxis group, comparing with one-day prophylaxis group (5.6% vs. 7.1%). Deep and organ/space SSIs incidence rates were 3-fold lower than superficial SSIS, without differences between groups, probably due to a very few numbers of these infections.

It was confirmed in the other studies that superficial SSIs are more frequent than deep infections [14], while the latter are often more serious. Studying the risk of complications after colorectal surgery, Kamboj and colleagues revealed that superficial incisional SSIs were four times more common than deep incisional infections [15]. Antibiotic prophylaxis can generally reduce the overall incidence of SSIs, but the specific type of infection can be influenced by factors related to the location and severity of the infection. The optimal duration for prophylactic antimicrobial usage is still varied despite the World Health Organization [4] and the Centers for Disease Control and Prevention [16] recommendation of a single dose before surgery without additional doses, within 60 to 120 min before skin incision, considering half-life of antibiotics, according WHO, or two half-lives of the drug, according the American Society of Health-System Pharmacists—ASHF [4,5]. For obese patients, the pharmacokinetics of medications might differ; therefore, adjusting dosages relative to body weight could be suitable for these individuals. The relative benefits of intraoperative re-dose in diabetic patients were observed too [17]. In our study, the one-day prophylaxis group consisted of patients significantly overweight and obese, a higher percentage of whom had an ASA score 3–4, a higher number of drains, longer duration of drain use, and a higher percentage of them were hospitalized in the ICU. Therefore, extended prophylaxis may be protective for such patients. Though considering the beneficial role of extended antibiotic prophylaxis in reducing superficial SSIs, it is not sufficient to omit the recommendation for one-day prophylaxis. To more fully examine the difference in the occurrence of all HAIs, SSIs and UTI after the use of prophylaxis for 1 day and extended prophylaxis, we performed a PSM analysis. This analysis allow to control of confounding factors between the two groups. Before PSM, using the traditional regression analysis, extended antibiotic prophylaxis (>1 day) was independently associated with a significantly lower risk of HAIs. Using the PSM, the difference of this association was no longer statistically significant. It is obvious that the association between one-dose prophylaxis was distorted by the influence of confounding variables.

Taking into account the small number of deep/organ space SSIs, a study with at least three times higher sample size of patients is needed to be able to recommend personalized prophylaxis for patients at high risk of infection. This potential study could be a follow-up study that, besides demographic characteristics, could include different preoperative data with which risk of infection could be estimate before operation, such as BMI data, comorbidities especially calculated via the Charleston index, data of previous colon surgery, glucose level, and immune status. Furthermore, it could be a randomized study to test the effect of personalized therapy, given that randomization is a powerful tool for mitigating confounding [18].

One of the adverse events of prolonged antibiotic use is *Clostridioides difficile* infections (CDIs). Advanced age, previous hospitalization, proton pump, and antibiotic use are well-known risk factors in developing a CDI postoperatively [19]. It was estimated that a combination of factors, such as those already mentioned, but also gut microbiota dysbiosis, prolonged exposure to chemotherapy, the role of bile acids in colon carcinogenesis, and colorectal surgery itself increase the risk of CDI in colorectal cancer patients [20,21]. A higher risk of CDI following prolonged consumption of antibiotic prophylaxis has been reported [22]. We did not find the difference regarding CDI occurrence in the one-day and extended antibiotic prophylaxis group of colorectal cancer surgery patients. This result may be due to the small number of patients with *C. difficile*. It is known that a small sample size analysis can lead to misinterpretations of data and inaccurate conclusions [23]. The cause reason for the low number of *C. difficile* may probably be underdiagnosis due to limited awareness among healthcare professionals of this type of infection or the complexity of accurate laboratory testing [24]. Additionally, the failures in diagnostics after discharge from the hospital may lead to the under-registration of diarrhea. Our patients had a relatively short hospital stay after operation, an average of 4 days, without significant differences between groups. Before being discharged from the hospital, patients were instructed to pay attention to the possible occurrence of diarrhea and to report symptoms to their healthcare provider. One month after discharge, all patients in our study were contacted directly during control medical examination by one of the authors (VN) to obtain information about quality of life and about symptoms of diarrhea or proven infection with *C. difficile*. Those who did not come to the examination were contacted by phone. We believe that there was no under-registration after discharge from the hospital. Nevertheless, a prospective study in which extensive surveillance of the occurrence of diarrhea and timely collection of samples for *C. difficile* and its toxins would provide a realistic picture of the frequency of *C. difficile* among patients who underwent colorectal surgery in our hospital.

In colorectal surgery, the first-generation cephalosporin, cefazolin, is recommended based on its pharmacokinetic profile and activity against staphylococci [5,25]. In addition, metronidazole is commonly combined with other antibiotics due to its effectiveness against aerobic bacteria and their prevalence in the colon and rectum [26].

All patients received (oral/IV) metronidazole besides the first generation of cephalosporins in approximately 97% of cases, which is in line with international [16] and national recommendations for antibiotic prophylaxis in colorectal surgery [27].

Urinary tract infections (UTIs) were the second most common localization of HAIs, with a cumulative incidence of 6.3%, without differences between groups in our study. Colorectal surgery, particularly rectal resections, is associated with a higher risk of UTIs compared to other types of gastrointestinal or non-gastrointestinal surgeries. The association between contaminated/dirty wounds and UTIs suggests that local contamination is strongly associated with UTIs [28].

We revealed that all bloodstream infections occurred in the extended prophylaxis group. BSIs involve patients immediately after surgical operations and are more common among patients over 65 years old, with an ASA score > 2, and also with comorbidities such as diabetes and cardiovascular disease [29]. They can decrease the survival rate of patients and lead to deep vein thrombosis. The most frequent sources of a BSI are anastomotic leakage, intravascular catheters, UTI, pseudomembranous colitis, and postoperative ileus [29,30]. All BCIs in our patients were primary, without an obvious origin. However, they needed prompt treatment, taking into account that they can be associated with a death rate similar to that seen with secondary BSIs.

Less than 50% of surgical site infections (SSIs) were laboratory confirmed, which is in contrast to findings in other developed countries where this percentage is much higher [10]. In a similar study in a tertiary center, all SSIs had a wound culture taken at the time of infection [31]. SSIs after colorectal surgery are often caused by bacteria originating from the patient’s intestinal flora. However, SSIs can be diagnosed with negative cultures in 10–30% of cases, even when the infection is clinically apparent probably due to prior antimicrobial therapy [32]. In our study, Gram-negative bacteria accounted for about 54% of the isolates, similar to results from other studies [10]. The high proportion of *Enterococcus* spp. may be explained by the specificities of the antibiotic protocol. Namely, it is known that the use of cephalosporins and metronidazole in antibiotic prophylaxis in colorectal surgery, which is in accordance with the recommendations for antibiotic prophylaxis in colorectal surgery [33], has limited activity against *Enterococcus* spp., which are normal inhabitants of the digestive tract. It is important to note that slightly more than one-third of the isolates were resistant to antibiotics, and no pathogen was pan-resistant. This indicates that the use of antibiotics in our hospital was not abused. It is well known that the irrational use of antibiotics can significantly contribute to the development of antibiotic resistance, which makes it difficult to effectively treat infections. The predominance of Gram-negative bacilli in the etiology of UTIs in our population was observed. Most UTIs are due to the colonization of the urogenital tract with rectal and perineal flora. *Candida* spp., a fungal pathogen, was the causative agent in 14.3% of our cases, suggesting possible immunocompromise or prior antibiotic therapy in these patients. The UTIs following colorectal surgery may have a higher percentage of resistant pathogens compared to surgical site infections (SSIs) in the same setting. This is due to factors like the inherent complexity of UTIs and the potential for antibiotic misuse in both treating and preventing these infections [28]. BSIs are often caused by Gram-positive and Gram-negative bacteria after colorectal surgery. The common sources of sepsis in patients who undergo colorectal surgery are SSIs, anastomotic leaks, and other intra-abdominal infections [29]. *Enterococcus* spp. and Enterobacteriaceae were the causative agents of BSIs in our patients.

The main limitation is that our study was conducted in a single center. However, our hospital is the largest university hospital in the country, to which not only residents of the capital but also those from all over the country, referred either from regional general hospitals or from primary health care centers, gravitate. Namely, the social security fund covers treatment in a tertiary institution if the regional hospitals lack adequate resources. Additionally, it should be noted that health insurance in Serbia is available to all categories of the population with adequate documentation. Potential study bias can be the overestimation of the severity of illness because only more severe patients are treated in tertiary healthcare hospitals. This is reflected in more frequent and longer ICU hospitalizations and a greater number of drains being worn for longer periods, and consequently, higher rates of all HAIs and SSIs.

## 4. Materials and Methods

The prospective cohort study was conducted at the Clinic for Digestive Surgery, University Clinical Center of Serbia. The study included adult patients who underwent elective colorectal surgery between June 2022 and December 2023.

Eligible participants were patients aged 18 years or older scheduled for elective colorectal surgery. Exclusion criteria included emergency procedures, surgeries for inflammatory bowel disease, metastatic colorectal cancer, and incomplete documentation on antibiotic use.

Patients were categorized into two groups based on the duration of antibiotic prophylaxis. The one-day group included patients who received a prophylactic antibiotic dose only on the day of surgery, without continuation postoperatively. All patients received the initial prophylactic dose within 60 min prior to surgical incision, in accordance with institutional protocols and international guidelines. The extended prophylaxis group included patients whose antibiotic usage was continued beyond the day of surgery.

Clinical and demographic data were collected using a standardized form. Variables included age, sex, body mass index, smoking and alcohol use, Charlson Comorbidity Index, ASA physical status classification, prior chemotherapy or radiotherapy, wound classification, surgical type (colon, rectum, or both), intensive care unit admission, number of surgical drains, duration of drain use, number of staff present in the operating room, and length of hospitalization before surgery. The class of antibiotic used and total duration of prophylaxis were also recorded.

The primary outcome was the occurrence of healthcare-associated infections, including:

Surgical site infections (SSIs) (superficial, deep, and organ-space).

Urinary tract infections (UTIs).

Bloodstream infections (BSIs).

*Clostridioides difficile* infections (CDIs).

Pressure ulcer infections.

Infections were diagnosed according to the European Centre for Disease Prevention and Control (ECDC) definitions, adapted to national guidelines [34].

The study was conducted in accordance with the principles of the Declaration of Helsinki and was approved by the Ethics Committee of the University Clinical Center of Serbia (Protocol No. 808/5). All participants provided written informed consent prior to participation.

The sample size was calculated using Slovin’s formula:$$n=\frac{N}{1+N\times e^{2}}$$
where *N* represents the estimated number of patients who underwent colorectal surgery at the Clinic for Digestive Surgery—First Surgical Clinic of the University Clinical Center of Serbia during 2019 and 2021, and *e* is the margin of error set at 5% (0.05). Data from 2020 were excluded due to the onset of the COVID-19 pandemic and its impact on hospital capacity and surgical volume. Based on a total of 621 patients operated during the reference years, the minimum required sample size was calculated to be 243 participants.

Out of 300 patients initially enrolled in the study, 22 were excluded because they underwent a minimally invasive (laparoscopic) procedure, which differs significantly from open surgery in terms of infection risk, length of hospital stay, and perioperative management. To maintain homogeneity of the study population, only patients who underwent open colorectal surgery were included in the final analysis (*n* = 278).

### Statistical Analysis

Continuous variables were summarized as mean ± standard deviation or median (range), and compared using the independent *t*-test or Mann–Whitney U test, as appropriate. The normality of data distribution was tested using Kolmogorov–Smirnov test. Categorical variables were expressed as frequencies and percentages and compared using the χ^2^ test or Fisher’s exact test. Multivariate binary logistic regression was performed to assess whether prolonged antibiotic prophylaxis (more than one day) was independently associated with the risk of HAIs, SSIs, and UTIs. The models were adjusted for age, sex, BMI, ASA score, Charlson comorbidity index, length of hospitalization before surgery, number of drains, duration of drain use, wound classification, ICU admission, and for UTI length of urinary catheter use. Adjusted risk ratios (RR) with 95% confidence intervals (CIs) were reported, and a *p*-value < 0.05 was considered statistically significant. The enter method was used. Propensity score matching (PSM) was used to minimize confounding by indication between patients who received one-day and extended antibiotic prophylaxis. Propensity scores were estimated using a logistic regression model including age, sex, BMI, ASA score, Charlson comorbidity index, and ICU admission as covariates. Full matching was applied to optimally use all available data while creating weighted groups with similar propensity scores. Covariate balance before and after matching was assessed using standardized mean differences (SMDs), with values less than 0.1 considered indicative of good balance. Balance was further evaluated using variance ratios and empirical cumulative distribution function (eCDF) statistics. After matching, all covariates achieved substantial balance. Weighted logistic regression models were then fitted on the matched sample to estimate aRR and 95% CIs for HAIs, SSIs, and UTIs. All statistical analyses were performed using SPSS version 26.0 (IBM Corp., Armonk, NY, USA) and R 4.5.1. (R Core Team (2025), Vienna, Austria) within the RStudio environment (version 2025.05.1 Build 513; Posit Software, PBC, Boston, MA, USA. Available online: https://posit.co, accessed on 29 June 2025). R is a language and environment for statistical computing from the R Foundation for Statistical Computing, Vienna, Austria (available online: https://www.R-project.org/, accessed on 29 June 2025). The following R packages were used: MatchIt, survey, cobalt, tableone, and readxl.

## 5. Conclusions

One-day of prophylactic antibiotics may be sufficient in SSI prevention in patients undergoing elective colorectal surgery. The use of extended antibiotic prophylaxis beyond one day should be considered for high-risk patients at high risk of infection, particularly those requiring ICU care.

## Figures and Tables

**Table 1 antibiotics-14-00791-t001:** Demographic, clinical, and perioperative characteristics of patients according to the duration of antibiotic prophylaxis.

	Total *n* (%)	Antibiotic Prophylaxis for One Day *n* (%)	Antibiotic Prophylaxis for More than One Day *n* (%)	*p* Value
Sex				
Male	155 (55.8)	53 (53.5)	102 (57.0)	0.579
Female	123 (44.2)	46 (46.5)	77 (43.0)	
Age—mean ± SD	65.0 ± 10.1	64.3 ± 11.5	65.4 ± 9.3	0.386
Charlson comorbidity index—median (min–max)	4 (2–8)	4 (2–7)	4 (2–8)	0.771
ASA				
1	11 (4.0)	5 (5.2)	6 (3.4)	0.233
2	126 (45.7)	51 (52.6)	75 (41.9)	
3	134 (48.6)	39 (40.2)	95 (53.1)	
4	5 (1.8)	2 (2.1)	3 (1.7)	
ASA				
1–2	137 (49.6)	56 (57.7)	81 (45.3)	**0.048**
3–4	139 (50.4)	41 (42.3)	98 (54.7)	
Wound classification				
Clean-contaminated	273 (98.2)	98 (99.0)	175 (97.8)	0.684
Contaminated	4 (1.4)	1 (1.0)	3 (1.7)	
Dirty	1 (0.4)	0 (0.0)	1 (0.6)	
Smoking status				
Current smoker	65 (23.4)	18 (18.2)	47 (26.3)	0.242
Former smoker	96 (34.5)	34 (34.3)	62 (34.6)	
Nonsmoker	117 (42.1)	47 (47.5)	70 (39.1)	
Pack-years	30.0 (0.3–147.0)	23.7 (0.7–147.0)	30.0 (0.3–129.0)	0.441
Alcohol consumption	138 (49.6)	45 (45.5)	93 (52.0)	0.299
BMI				
Underweight	13 (4.7)	3 (3.0)	10 (5.6)	**0.021**
Normal weight	114 (41.0)	45 (45.5)	69 (38.5)	
Overweight	98 (35.3)	41 (41.4)	57 (31.8)	
Obesity	53 (19.1)	10 (10.1)	43 (24.0)	
Radiotherapy	67 (24.1)	23 (23.2)	44 (24.6)	0.801
Chemotherapy	69 (24.8)	25 (25.3)	44 (24.6)	
Number of staff in the operating room—median (min–max)	5 (4–8)	5 (4–7)	5 (4–8)	0.771
Number of drains—median (min–max)	1 (1–5)	1 (1–4)	1 (1–5)	**<0.001**
Duration of drain use (days)—median (min–max)	4 (1–32)	3 (1–10)	4 (2–32)	**<0.001**
Length of urinary catheter use (days)—median (min–max)	3 (1–65)	2 (1–44)	4 (1–65)	**<0.001**
ICU admission	125 (45.0)	16 (16.2)	109 (60.9)	**<0.001**
ICU stay—median (min–max)	1 (1–40)	1 (1–14)	1 (1–40)	0.343
Length of hospitalization before surgery (days)—median (min–max)	4 (0–31)	4 (1–31)	4 (0–21)	0.078

SD—Standard deviation; ASA—American Society of Anesthesiology; BMI—Body mass index; ICU—Intensive care unit.

**Table 2 antibiotics-14-00791-t002:** Incidence of healthcare-associated infections according to the duration of antibiotic prophylaxis.

	Total *n* (%)	Antibiotic Prophylaxis for One Day *n* (%)	Antibiotic Prophylaxis for More than One Day *n* (%)	*p* Value
HAI	47 (16.9)	17 (17.2)	30 (16.8)	0.930
SSI	27 (9.7)	11 (11.1)	16 (8.9)	0.558
UTI	19 (6.8)	6 (6.1)	13 (7.3)	0.704
BSI	7 (2.5)	0 (0.0)	7 (3.9)	0.053
CDI	3 (1.1)	2 (2.0)	1 (0.6)	0.289
Decubital ulcer infections	2 (0.7)	1 (1.0)	1 (0.6)	1.000

HAI—Healthcare-associated infection; SSI—Surgical site infection; UTI—Urinary tract infection; BSI—Bloodstream infection; CDI—*Clostridioides difficile* infection.

**Table 3 antibiotics-14-00791-t003:** Multivariate binary logistic regression model for predicting healthcare-associated infections before and after propensity score matching.

	Traditional Regression Analysis			Propensity Score Matching		
	aRR	95% CI	*p* Value	aRR	95% CI	*p* Value
Antibiotic prophylaxis for more than one day	**0.33**	**0.14–0.78**	**0.012**	0.81	0.33–1.98	0.636
Age	0.99	0.94–1.06	0.888	0.96	0.89–1.03	0.260
Sex						
Male						
Female	1.11	0.55–2.22	0.774	0.84	0.36–1.95	0.685
BMI	0.98	0.91–1.05	0.603	0.93	0.86–1.01	0.105
ASA	1.34	0.70–2.58	0.371	**2.54**	**1.08–6.01**	**0.033**
Charlson comorbidity index	1.22	0.74–2.02	0.441	1.60	0.86–2.95	0.134
Length of hospitalization before surgery	1.07	0.99–1.15	0.056	1.03	0.95–1.12	0.435
Number of drains	1.18	0.72–1.92	0.514	1.12	0.61–2.07	0.720
Duration of drain use	1.08	0.97–1.20	0.180	**1.11**	**1.00–1.22**	**0.046**
Wound classification						
Clean-contaminated	Ref.			Ref.		
Contaminated	6.72	0.53–84.42	0.140	**22.0**	**3.78–128.0**	**<0.001**
Dirty	NA			NA		
ICU admission	**4.76**	**2.07–10.95**	**<0.001**	**4.51**	**1.74–11.69**	**0.002**

aRR—Adjusted risk ratio; CI—Confidence interval; BMI—Body mass index; ICU—Intensive care unit; NA—Not applicable due to small number of cases in this category.

**Table 4 antibiotics-14-00791-t004:** Multivariate binary logistic regression model for predicting SSI before and after propensity score matching.

	Traditional Regression Analysis			Propensity Score Matching		
	aRR	95% CI	*p* Value	aRR	95% CI	*p* Value
Antibiotic prophylaxis for more than one day	**0.26**	**0.09–0.75**	**0.013**	0.48	0.17–1.36	0.167
Age	0.97	0.90–1.04	0.419	0.94	0.85–1.02	0.142
Sex						
Male						
Female	1.46	0.60–3.54	0.405	0.88	0.30–2.60	0.813
BMI	1.0	0.91–1.10	0.978	0.97	0.88–1.08	0.628
ASA	2.14	0.89–5.14	0.088	**3.61**	**1.26–10.32**	**0.017**
Charlson comorbidity index	1.35	0.74–2.46	0.326	1.82	0.86–3.84	0.116
Length of hospitalization before surgery	1.09	0.99–1.19	0.052	1.04	0.94–1.15	0.473
Number of drains	1.01	0.53–1.89	0.989	0.95	0.38–2.42	0.921
Duration of drain use	1.09	0.96–1.24	0.163	**1.14**	**1.01–1.28**	**0.030**
Wound classification						
Clean-contaminated	Ref.					
Contaminated	19.74	0.97–399.60	0.052	4.68	0.18–122.3	0.353
Dirty	NA			NA		
ICU admission	**5.63**	**1.91–16.58**	**0.002**	**5.34**	**1.70–16.83**	**0.004**

aRR—Adjusted risk ratio; CI—Confidence interval; BMI—Body mass index; ICU—Intensive care unit; NA—Not applicable due to small number of cases in this category.

**Table 5 antibiotics-14-00791-t005:** Multivariate binary logistic regression model for predicting UTIs before and after propensity score matching.

	Traditional Regression Analysis			Propensity Score Matching		
	aRR	95% CI	*p* Value	aRR	95% CI	*p* Value
Antibiotic prophylaxis for more than one day	0.71	0.19–2.66	0.612	2.76	0.52–14.70	0.233
Age	1.02	0.92–1.13	0.738	0.995	0.90–1.09	0.924
Sex						
Male						
Female	0.79	0.25–2.44	0.677	0.60	0.18–1.98	0.403
BMI	1.06	0.95–1.19	0.311	1.07	0.94–1.21	0.293
ASA	0.76	0.27–2.10	0.598	1.07	0.39–2.92	0.901
Charlson comorbidity index	0.87	0.36–2.08	0.755	0.88	0.39–1.99	0.764
Length of hospitalization before surgery	0.98	0.87–1.11	0.785	0.95	0.85–1.06	0.363
Number of drains	1.34	0.63–2.87	0.445	1.12	0.44–2.83	0.813
Duration of drain use	0.88	0.67–1.16	0.358	0.88	0.66–1.17	0.370
Wound classification						
Clean-contaminated	Ref.					
Contaminated	2.81	0.19–42.31	0.455	**16.2**	**3.69–71.36**	**<0.001**
Dirty	NA			NA		
ICU admission	1.98	0.55–7.18	0.299	2.20	0.41–11.88	0.359
Length of urinary catheter use	**1.10**	**1.04–1.16**	**<0.001**	**1.11**	**1.03–1.20**	**0.005**

aRR—Adjusted risk ratio; CI—Confidence interval; BMI—Body mass index; ICU—Intensive care unit; NA—Not applicable due to small number of cases in this category.

**Table 6 antibiotics-14-00791-t006:** Resistance of causative agents of SSIs, UTIs, and BSIs.

	GLY		C3G		CAR		PDR
SSI	R	S	R	S	R	S	No
*Enterococcus* spp.	1 (16.7)	5 (83.3)					6 (100.0)
*Klebsiella* spp.			3 (100.0)		1 (33.3)	2 (66.7)	3 (100.0)
*Escherichia coli*				2 (100.0)		2 (100.0)	2 (100.0)
*Pseudomonas aeruginosa*					1 (50.0)	1 (50.0)	2 (100.0)
UTI							
*Klebsiella* spp.			5 (83.3)	1 (16.7)	2 (33.3)	4 (66.7)	6 (100.0)
*Enterococcus* spp.	1 (33.3)	2 (66.7)					3 (100.0)
*Candida* spp.							
*Klebsiella* spp.			1 (50.0)	1 (50.0)	2 (100.0)		2 (100.0)
*Acinetobacter* spp.					1 (100.0)		1 (100.0)
*Pseudomonas aeruginosa*						1 (100.0)	1 (100.0)
*Proteus mirabilis*				1 (100.0)		1 (100.0)	1 (100.0)
BSI							
*Enterococcus faecium*		2 (100.0)					2 (100.0)
*Pseudomonas aeruginosa*						2 (100.0)	2 (100.0)
*Klebsiella pneumoniae*			1 (50.0)	1 (50.0)	1 (50.0)	1 (50.0)	2 (100.0)

Data presented as number (%); GLY—glycopeptides; C3G—third-generation cephalosporins; CAR—carbapenems; PDR—pandrug-resistant (resistant to all antibiotics).

## Data Availability

The data presented in this study are available upon request from the corresponding author due to privacy reasons.

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
