# Peer review of "Impact of Antibiotic Prophylaxis Duration on the Incidence of Healthcare-Associated Infections in Elective Colorectal Surgery"

_antibiotics, 2025, doi:10.3390/antibiotics14080791_

Round 1
Reviewer 1 Report
Comments and Suggestions for Authors
Dear authors,
I appreciate the efforts put in by the authors to investigate the association between antibiotic prophylaxis duration and healthcare-associated infections (HAIs) in elective open colorectal surgery. The strengths of the study, what I found, are;
- Well-defined study design
- Adequate Sample Size
- Real-World Relevance: Addresses deviations from WHO/CDC guidelines in practice.
- Multivariate Analysis: Adjusts for confounders like ICU admission, ASA score, BMI, etc.
- Clear Findings: Extended prophylaxis was associated with reduced superficial SSIs and overall SSIs.
However, certain data needs to be clear or elaborated for a better understanding of the study results.
-
Risk of Confounding by Indication: Sicker patients received extended prophylaxis. Consider Propensity Score Matching to reduce confounding.
- The study concludes that extended prophylaxis reduces SSIs and HAIs, which conflicts with current WHO/CDC recommendations against prolonged use due to antimicrobial resistance risks. This needs more nuanced discussion and possible exploration of patient subgroups (e.g., high-risk ICU patients).
- The finding of no difference in CDI rates despite extended prophylaxis deserves more critical analysis, as it contradicts some earlier studies.
- No Antimicrobial Resistance Data: A key omission given concerns of overuse. Include the resistance data if available.
- Some sections (especially Results and Discussion) could benefit from better English language and clearer flow to avoid repetition.
Author Response
Reviwer 1
We are grateful to the reviewer for his valuable suggestions. paper. We have tried to respond to all the comments as best as possible and believe that the quality of the manuscript has improved.
We'd like to thank the reviewers for their
- Risk of Confounding by Indication: Sicker patients received extended prophylaxis. Consider Propensity Score Matching to reduce confounding.
Response: We thank to this valuable remark. We did the Propensity Score Matching Analysis and explained it in the Method section. We added results in the Tables 3,4 and 5. Indeed, contrary to the traditional logistic regression, after adjusting for potential confounding with the PSM analysis, the association of extended prophylaxis and lower SSIs incidence are not statistically significant. Also, we added the paragraph in the Discussion section.
- The study concludes that extended prophylaxis reduces SSIs and HAIs, which conflicts with current WHO/CDC recommendations against prolonged use due to antimicrobial resistance risks. This needs more nuanced discussion and possible exploration of patient subgroups (e.g., high-risk ICU patients).
Response: In accordance with the previous reviewer's comment, we performed a PSM. The results obtained are in accordance with the WHO and CDC recommendations. We changed the conclusion.
- The finding of no difference in CDI rates despite extended prophylaxis deserves more critical analysis, as it contradicts some earlier studies.
Response: We thank to this important remark. We added one paragraph with possible explanation of our results concerning CDI in the Discussion section.
- No Antimicrobial Resistance Data: A key omission given concerns of overuse. Include the resistance data if available.
Response: Thank you for this suggestion. We acknowledge the importance of antimicrobial resistance data and added them to a new table 6, and comments in the Discussion.
Comments on the Quality of English Language
Some sections (especially Results and Discussion) could benefit from better English language and clearer flow to avoid repetition.
Response: The paper has been carefully revised by a native English speaker to improve the grammar and readability.
Reviwer 1
We are grateful to the reviewer for his valuable suggestions. paper. We have tried to respond to all the comments as best as possible and believe that the quality of the manuscript has improved.
We'd like to thank the reviewers for their
- Risk of Confounding by Indication: Sicker patients received extended prophylaxis. Consider Propensity Score Matching to reduce confounding.
Response: We thank to this valuable remark. We did the Propensity Score Matching Analysis and explained it in the Method section. We added results in the Tables 3,4 and 5. Indeed, contrary to the traditional logistic regression, after adjusting for potential confounding with the PSM analysis, the association of extended prophylaxis and lower SSIs incidence are not statistically significant. Also, we added the paragraph in the Discussion section.
- The study concludes that extended prophylaxis reduces SSIs and HAIs, which conflicts with current WHO/CDC recommendations against prolonged use due to antimicrobial resistance risks. This needs more nuanced discussion and possible exploration of patient subgroups (e.g., high-risk ICU patients).
Response: In accordance with the previous reviewer's comment, we performed a PSM. The results obtained are in accordance with the WHO and CDC recommendations. We changed the conclusion.
- The finding of no difference in CDI rates despite extended prophylaxis deserves more critical analysis, as it contradicts some earlier studies.
Response: We thank to this important remark. We added one paragraph with possible explanation of our results concerning CDI in the Discussion section.
- No Antimicrobial Resistance Data: A key omission given concerns of overuse. Include the resistance data if available.
Response: Thank you for this suggestion. We acknowledge the importance of antimicrobial resistance data and added them to a new table 6, and comments in the Discussion.
Comments on the Quality of English Language
Some sections (especially Results and Discussion) could benefit from better English language and clearer flow to avoid repetition.
Response: The paper has been carefully revised by a native English speaker to improve the grammar and readability.
Reviewer 2 Report
Comments and Suggestions for Authors
This prospective cohort study investigates the effect of the duration of antibiotic prophylaxis on the incidence of healthcare-associated infections (HAIs) in patients undergoing elective colorectal surgery. The research is timely and clinically relevant, given the ongoing debate surrounding the optimal duration of prophylactic antibiotic administration in surgical settings.
The strengths of the study lie in its prospective design and the clear stratification of patients based on the duration of antibiotic administration. The inclusion of multivariate analysis strengthens the findings by adjusting for potential confounders, revealing that prolonged prophylaxis was independently associated with a significantly lower risk of HAIs (RR = 0.30, p = 0.010) and surgical site infections (SSIs) (RR = 0.24, p = 0.011). The outcome measures are well defined and the antibiotic regimens described are in line with common clinical practice, with most patients receiving first-generation cephalosporins plus metronidazole.
This study provides evidence that extended antibiotic prophylaxis can reduce HAIs and SSIs in elective colorectal surgery.
The study demonstrates a well-designed and comprehensive approach. The paper demonstrates a well-written and meticulously executed study with a well-designed methodology. The bibliography reflects up-to-date references, contributing to the credibility of the study and demonstrating a thorough review of the existing literature.
Overall, the paper is commendable for its clarity, comprehensive approach, and adherence to rigorous scientific methods. The figures, tables, and photos in the paper are well-prepared, contributing to the overall clarity and effectiveness of the study.
The study holds significant interest for a diverse audience, including scientists, industrial professionals, and diagnostic companies.
Quality of English language is fine.
I recommend accepting the publication of this paper in its present form
Author Response
We thank the reviewer for all the comments.
Reviewer 3 Report
Comments and Suggestions for Authors
The aim of this study was to determine whether the duration of antibiotic prophylaxis influences the incidence of nosocomial infections (HAIs) in patients undergoing colorectal surgery. The study was designed as a prospective cohort study, with a total of 278 selected adult patients from a single center (University Clinical Center of Serbia). Two groups were examined (one-day and more than one-day antibiotic prophylaxis). Demographic data, clinical characteristics, perioperative variables, and infection outcomes were collected and analyzed. Multivariate binary logistic regression was used in addition to statistical comparison of the variables. The authors found that prophylaxis of more than one day is associated with a significantly lower risk of nosocomial infections. The authors conclude from the results that prolonged prophylaxis may be beneficial in selected high-risk patients, especially those requiring intensive care.
The main problem with the study appears to be the fact that the patients came from only a single observed center, but the resulting limitations are adequately discussed in the article. For more detailed information (e.g., regarding personalized prophylaxis for patients at high risk of infection), a significantly larger number of patients would be necessary, and the authors state that this number is three times the current number.
From the reviewers perspective, the study appears well-structured, and the limitations are adequately stated and discussed. Conducting a follow-up study with the inclusion of 800 to 1000 patients seems sensible to achieve greater significance. As a small suggestion for improvement, the discussion should specify more precisely which variables should be examined in this potential follow-up study.
Line 34-36: Please explain the sentence "The overall incidence of HAIs was 16.9%, with no significant difference between patients receiving one-day versus extended antibiotic Prophylaxis"! Was there really no significant difference, ore is this some kind of spelling error?
Author Response
Reviwer 3
The aim of this study was to determine whether the duration of antibiotic prophylaxis influences the incidence of nosocomial infections (HAIs) in patients undergoing colorectal surgery. The study was designed as a prospective cohort study, with a total of 278 selected adult patients from a single center (University Clinical Center of Serbia). Two groups were examined (one-day and more than one-day antibiotic prophylaxis). Demographic data, clinical characteristics, perioperative variables, and infection outcomes were collected and analyzed. Multivariate binary logistic regression was used in addition to statistical comparison of the variables. The authors found that prophylaxis of more than one day is associated with a significantly lower risk of nosocomial infections. The authors conclude from the results that prolonged prophylaxis may be beneficial in selected high-risk patients, especially those requiring intensive care.
The main problem with the study appears to be the fact that the patients came from only a single observed center, but the resulting limitations are adequately discussed in the article. For more detailed information (e.g., regarding personalized prophylaxis for patients at high risk of infection), a significantly larger number of patients would be necessary, and the authors state that this number is three times the current number.
From the reviewers perspective, the study appears well-structured, and the limitations are adequately stated and discussed. Conducting a follow-up study with the inclusion of 800 to 1000 patients seems sensible to achieve greater significance. As a small suggestion for improvement, the discussion should specify more precisely which variables should be examined in this potential follow-up study.
Response: We thank the reviewer for all comments.
We added in the discussion that potential follow-up study could include besides demographic characteristics, different preoperative data with which risk of infection could be estimate before operation, such as: and data BMI, coomorbidites, especially calculated via Charleston index, data of previous colon surgery, glucose level, immune status. In addition, it could be recommended to conduct a randomized study to test the effect of personalized therapy, given that randomization is a powerful tool for mitigating confounding.
Line 34-36: Please explain the sentence "The overall incidence of HAIs was 16.9%, with no significant difference between patients receiving one-day versus extended antibiotic Prophylaxis"! Was there really no significant difference, or is this some kind of spelling error?
Response: We put in the parenthesis that all HAIs incidence in one-day group was 17.2 and 16.8 in the extended prophylaxis group which is not statistical significant (p= 0.930).
Round 2
Reviewer 1 Report
Comments and Suggestions for Authors
Dear authors,
Thanks for making the necessary changes to the manuscript as per the suggestions.